# Comparative specificity and sensitivity of NS1-based serological assays for the detection of flavivirus immune response

**Erick Mora-Cárdenas**[1], **Chiara Aloise**[1], **Valentina Faoro**[1], **Nataša Knap Gašper**[2], **Miša Korva**[2], **Ilaria Caracciolo**[3], **Pierlanfranco D'Agaro**[3], **Tatjana Avšič-Županc**[2], **Alessandro Marcello**[1] *

**1** Laboratory of Molecular Virology, International Centre for Genetic Engineering and Biotechnology (ICGEB), Trieste, Italy, **2** Laboratory of Diagnostics of Zoonoses and WHO Centre, Institute of Microbiology and Immunology, Faculty of Medicine, University of Ljubljana, Ljubljana, Slovenia, **3** Regional reference Centre for Arbovirus infections, Department of Medical, Surgical and Health Sciences, University of Trieste, Trieste, Italy

\* marcello@icgeb.org

**Data Availability Statement:** All relevant data are within the manuscript and its Supporting Information files.

## Abstract

Flaviviruses are relevant animal and human pathogens of increasing importance worldwide. The similarities of the initial clinical symptoms and the serological cross-reactivity of viral structural antigens make a laboratory diagnosis of flavivirus infection problematic. The main aim of the present study was the comparative specificity and sensitivity analysis of the non-structural protein NS1 as an antigen to detect flavivirus antibodies in sera from exposed individuals. A strategy for the purification of native recombinant non-structural protein 1 of representative flaviviruses including tick-borne encephalitis, West Nile, Zika and dengue virus was developed. The immunological properties of the purified antigens were analyzed using sera of immunized mice and of infected individuals in comparison with standard commercial assays. Recombinant NS1 protein was confirmed as a valuable option for the detection of flavivirus antibodies with reduced cross-reactivity and high sensitivity offering additional advantages for the detection of vaccine breakthrough cases.

## Author summary

Viruses transmitted by mosquitoes or ticks are on the rise worldwide and may cause life-threatening symptoms, such as brain damage or excessive bleeding. In recent years we experienced several epidemic threats caused by colonization of new geographic areas by the infected vector with large numbers of naive population at risk of infection. Furthermore, increased travel contributed to a rise of imported infections in non-endemic countries. Reliable diagnostics for these viruses have therefore become a pressing need. Serology of infection is complicated by a high level of cross reactivity among structural virion proteins that are generally used as antigens. Therefore, use of the viral non-structural protein NS1 secreted by Flaviviruses, a family of highly significant vector borne viruses that include that infectious cause of Zika, Dengue, west Nile fever and Tick-borne

**Funding:** Region Friuli Venezia Giulia (FVG) Project SEVARE and Beneficientia Stiftung Project BEN 2016/13 to AM; Slovenian Research Agency Project P3-0083 to TAZ. The funders had no role in study design, data collection and analysis, decision to publish, or preparation of the manuscript.

**Competing interests:** The authors have declared that no competing interests exist.

encephalitis, has been proposed as a more specific antigen for detecting specific immune responses. In this work NS1 from representative Flaviviruses were purified from mammalian cells to maintain full antigenic potential and used in serological assays in a comparative analysis with commercial assays. Results showed a high level of specificity and sensitivity of NS1-based assays that can be proposed in a variety of diagnostic formats, from standard assays to multiparametric portable devices for rapid deployment.

## Introduction

Flaviviruses are responsible for considerable human morbidity and mortality in areas where ecological, geographical and socioeconomic factors facilitate their propagation [1, 2]. Members of flavivirus can be delivered by ticks, such as Tick-borne encephalitis virus (TBEV), or mosquitoes such as West Nile virus (WNV), Zika virus (ZIKV), the four Dengue virus serotypes (DENV1-4), Yellow fever virus (YFV) and Japanese encephalitis virus (JEV) [3]. Traveling within endemic areas, rapid urbanization, widespread deforestation together with the possible adaptation of flavivirus to new habitats and host species have greatly contributed to the increase of flaviviral infections into previously non-endemic areas [2, 4]. Flaviviruses cause a range of symptoms that can lead to lethal neurological (TBEV, WNV) or hemorrhagic syndromes (DENV, YFV) [5].

Virus isolation, detection of the viral RNA genome and detection of viral antigens for laboratory diagnosis of flavivirus infection are highly specific but limited to the short window of time (2–3 weeks) of the viremic phase [6, 7]. IgM antibodies can be detected in most patients by day 3–5 after the onset of symptoms and remain for several months following the infection. IgG antibodies are detected at low titer a couple of days after the IgM and increase slowly. IgG antibodies can be measured for many months and remain stable for decades [8]. In secondary infections, high levels of IgG antibodies that cross-react with many flaviviruses are detectable even in the acute phase and rise dramatically over the following two weeks [9]. Serological assays are widely used but suffer from broad antigenic cross-reactivity of anti-flavivirus antibodies [10, 11].

Flavivirus serology assays are generally based on structural antigens, while the viral non-structural protein 1 (NS1) has recently emerged as a valuable alternative [12–16]. NS1 is involved in various aspects of the virus lifecycle [17–19]. Most importantly, NS1 is secreted as a glycosylated hexamer in high amounts during the viremic phase and its detection is considered a marker of infection [20–22]. Anti-NS1 IgM appears early (2–3 days) following infection. Evaluation of acute infection by measuring anti-NS1 IgM antibodies has shown higher sensitivity compared to RT-PCR in sera samples of infected patients with dengue virus [6, 23]. Detection of IgG antibodies using purified NS1 antigens from different flaviviruses has been reported to show a low degree of cross-reactivity to related viruses and high reactivity to homologous NS1 antigens [12].

In the present study, a strategy is developed to produce and purify, from mammalian cells, fully antigenic recombinant NS1 proteins of representative flaviviruses endemic in Europe, such as TBEV and WNV, as well as tropical diseases, such as ZIKV and all DENV serotypes. These antigens are used to conduct a comparative analysis of their properties on sera from immunized mice and on well-characterized human samples. Results show high sensitivity and specificity compared to standard commercial assays both for IgM and IgG detection. An added value of this work is the demonstration that rNS1 serology has the unique ability to

detect TBEV vaccine breakthrough cases of infection, which is a problem of increasing importance in Europe.

## Results

### Expression and purification of flavivirus recombinant NS1 proteins

Expression plasmids encoding codon-optimized full-length nonstructural protein 1 (NS1) from TBEV, WNV, ZIKV and all DENV serotypes (DENV1-4) were generated. A Sec signal peptide at the N-terminus of the protein was included for optimal secretion and a 6x-histidine tag (6x-His) or V5 tag was also included at the C terminus immediately upstream of the stop codon for protein purification and mice immunization, respectively (Fig 1A). To ensure proper folding and glycosylation, plasmids encoding NS1 proteins were transiently transfected in human embryonic kidney 293T (HEK293T) cells. Culture supernatant containing secreted recombinant NS1 (rNS1) proteins was collected 48 hours post-transfection. Purifications of 6x-His-tagged rNS1 proteins present in the supernatant were carried out via nickel-affinity chromatography. Densitometric analysis of Coomassie-stained polyacrylamide gel electrophoresis (PAGE) showed between 87–92% purity for each purified protein (Fig 1B). The oligomeric status of the purified proteins was analyzed by WB using an antibody against the 6x-His-tag. WB of denatured purified rNS1 proteins diluted in reducing Laemmli sample buffer (LB) showed a single band for TBEV, meanwhile, for ZIKV and DENV1-4, a thicker band (overlapping of 2 or 3 bands depending on the glycosylation profile of NS1) was observed. In all cases, each band corresponds to the 50–55 kDa monomeric form of NS1 (Fig 1C, left panel). When the WB was performed in unheated/non-reducing conditions, mainly dimeric forms of NS1 proteins were observed. Interestingly, TBEV-NS1 proteins where uniquely stable as SDS-resistant hexameric species (Fig 1C, middle panel). Meanwhile, when the purified rNS1 proteins were analyzed by western blot under native conditions, they were composed of high molecular weight oligomers (Fig 1C, right panel). Although a more thorough biochemical analysis would be required to fully characterize the seven recombinants antigens, nevertheless these data are sufficient to establish the oligomeric nature of the secreted rNS1, compatible with the viral protein secreted by infected cells.

The glycosylation profile of purified rNS1 proteins was then assessed by digestion with endoglycosidase Hf (endo Hf), which removes high-mannose-content sugars, or peptide N-glycosidase F (PNGase F), which cleaves all sugar moieties from high mannose, hybrid, and complex oligosaccharides from N-linked glycoproteins. After treatment for 1.5 hours at 37˚C with Endo Hf, PNGase F or both enzymes, rNS1 proteins showed a shift in migration compared to untreated, as expected from a glycosylated protein (Fig 1D). In particular, all antigens, except for WNV rNS1, show a migration profile in the Endo H treated samples that is compatible with the pattern originally described for DENV NS1, with an Endo H resistant complex type glycan at N130 and an Endo H sensitive high mannose simple glycan at N207 [24]. The only variation to the theme is for WNV NS1 that shows complete digestion with Endo H (Fig 1D). This could be explained as the result of combining the efficient heterologous sec leader peptide and the RQ10NK mutation that confers high secretion efficiency [25]. Further analysis would be required to establish if this form of WNV rNS1, which displays mostly immature simple glycans, instead of Endo H resistant glycans typical of secreted WNV NS1 from infected cells, maintains intact immunological properties.

Together, these results confirm that rNS1 proteins purified from the supernatant of transfected mammalian HEK293T human cells maintain the oligomeric conformation characteristic of the secreted protein and are modified with N-linked glycans.

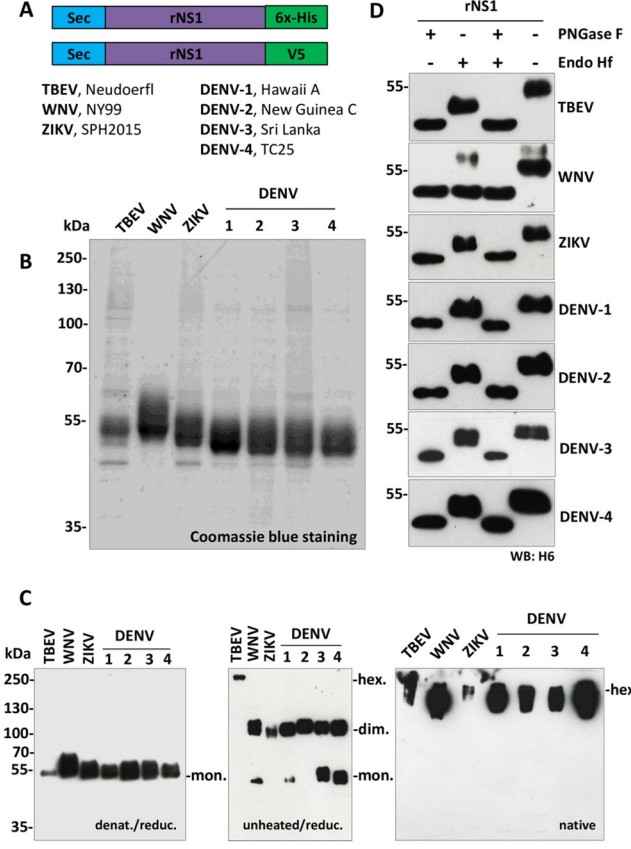

**Fig 1. Cloning, expression, and purification of rNS1.** A) Schematic representation of expression constructs pcDNA3.1-sec-NS1-6xHis (upper) and pcDNA3.1-sec-NS1-V5 (lower). Sec corresponds to an immunoglobulin leader sequence at the N-term; rNS1, recombinant nonstructural protein 1; 6xHis and V5 are tags cloned at the C-term for protein purification and mice immunization, respectively. Reference strains are also indicated. B) Coomassie blue staining of purified proteins: 2 μg of purified NS1 proteins were loaded for each virus. C) Western blot analysis of purified rNS1. Left panel, WB of purified rNS1 proteins diluted in reducing Laemmli sample buffer (LB) and denatured by boiling using an anti-6xHis-tag mAb (WB: H6). Middle panel, WB of rNS1 performed in non-denaturing/non-reducing conditions (without heating the proteins and LB without 2-βME). Right panel, WB of purified rNS1 proteins analyzed by western blot under native conditions (without heating the proteins; without 2-βME and without SDS in the LB buffer, gel, and running buffer). rNS1 monomers (mon), dimers (dim), and hexamers (hex) are indicated. D) Endoglycosidase analysis of purified rNS1. All rNS1 proteins were treated (+)/or not (-) with PNGase F and/or Endo Hf enzymes for 1.5 h at 37 ˚C. WB analysis was assessed by 10% SDS-PAGE under standard conditions using an anti-6xHis-tag mAb (WB: H6).

## Validation of rNS1 antigens by ELISA in immunized mice

Sera from immunized mice reactive for each rNS1 protein were used to check their immuno-logical properties. Expression and secretion of V5-tagged rNS1 proteins were checked in cell extract and culture supernatant of transiently transfected HEK293T cells (Fig 2A). All proteins showed good expression and secretion, with bands corresponding to their glycosylation status. Expression plasmids were therefore used for gene gun-mediated intradermal DNA immuniza-tion of BALB/c mice (4 mice each for TBEV, WNV, ZIKV, and DENV1-4). As an example, 4 different mice (Rx, Lx, RxLx, and unmarked) were immunized with TBEV-NS1 V5-tagged construct at days 1, 15, 22 and 29, and sera samples were collected at days 0 (pre-immune sera), 19 (bleeding I), 26 (bleeding II), 33 (bleeding III) and 46 (bleeding IV) (Fig 2B). Sera samples were used for IgM/IgG detection by rNS1-based ELISA. Cut-off values of P/N ratios were calculated using pre-immune sera (negative control) of all immunized mice. The optimal

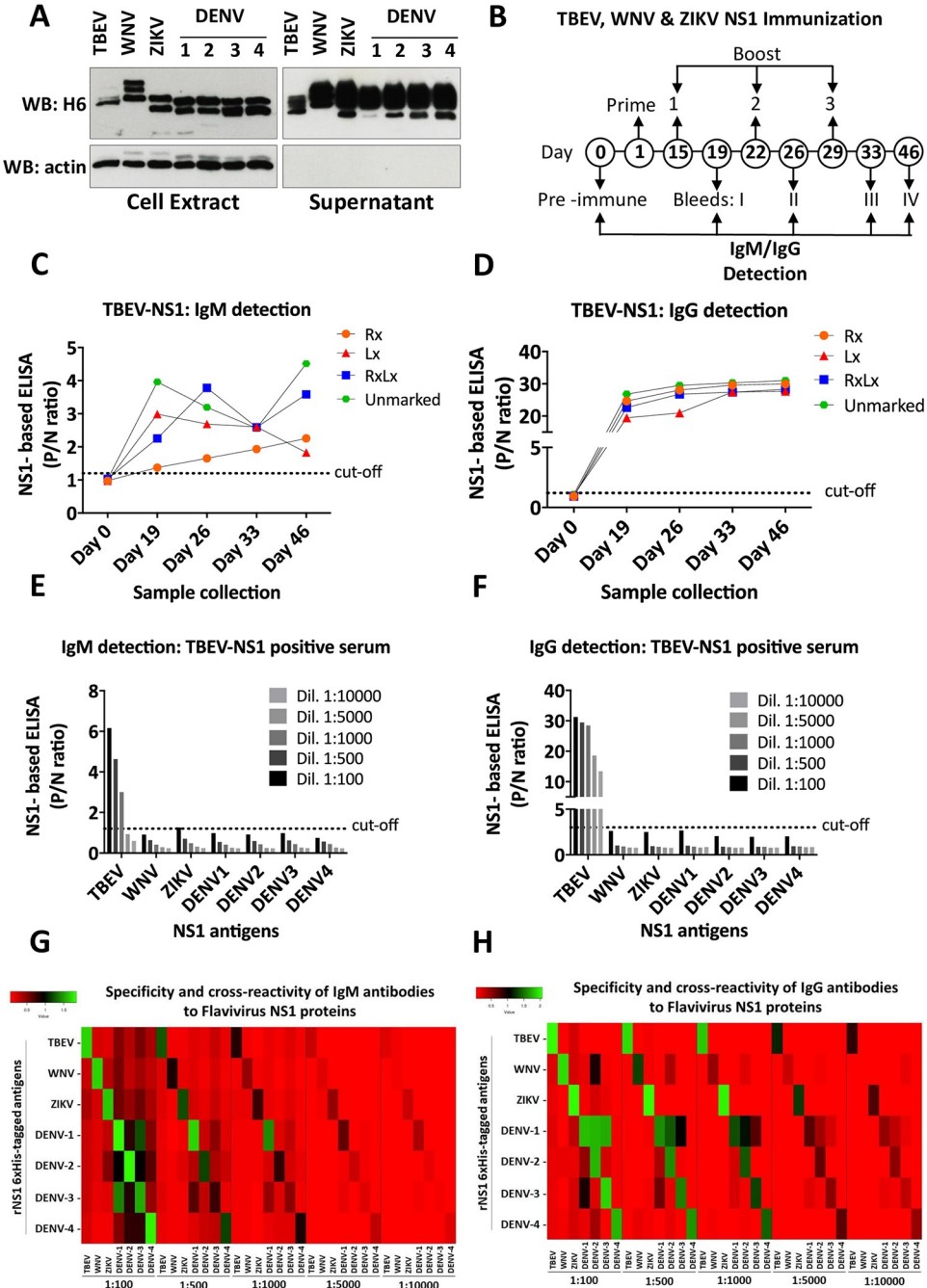

**Fig 2. Validation of rNS1 antigens by ELISA in immunized mice.** A) Expression and secretion of V5-tagged recombinant NS1 proteins of TBEV, WNV, ZIKV, and DENV1-4 in cell extract and supernatant of transfected HEK293T cells. Actin blots were included as a loading control for cell extracts and of clean supernatants. SDS PAGE was conducted in denaturing/reducing conditions. B) Schematic representation of the protocol followed for the immunization of BALB/c mice by gene gun technology. Numbers indicate the day when the prime-boost immunization was performed and the day of collection. C and D) TBEV rNS1 IgM/IgG ELISA with sera from four different mice immunized with TBEV-NS1 (mice were differentiated with a mark in the right ear (Rx), left (Lx), both ears (RxLx) or without the mark (unmarked)). Detection of IgM/IgG antibodies was performed at day 0, 19, 26, 33 and 46. E and F) TBEV, WNV, ZIKV, and DENV1-4 IgM/IgG rNS1 ELISA with sera from TBEV-NS1 immunized mice. Sera from day 46 post-immunization were diluted in blocking solution (2% milk in PBS) as indicated. G and H) Heatmaps of the IgM/IgG cross-reactivity. Several dilutions of each serum sample were analyzed in plates coated with homologous and heterologous rNS1 6xHis-tagged antigens. * each ELISA result includes the average of two biological replicates.

cut-off values for IgM and IgG detection fell at 1.2 and 3, respectively (S1 Table). IgM and IgG P/N ratios above the cut-off were obtained in all collected sera after the immunization (bleeding IV) with TBEV-NS1 (Fig 2C and 2D). P/N ratios for IgG detection were higher and more consistent between sera samples of the same group compared to IgM detection. Similar results were obtained for IgM and IgG detection in mice immunized with WNV and ZIKV NS1 V5-tagged constructs following the same immunization scheme as for TBEV (S1 Fig).

Since there is an increase in the number of reports showing that high levels of IgG can affect the detection of IgM antibodies [26, 27], we followed a different scheme for DENV1-4 NS1 immunization. Mice were immunized at fourteen days intervals and, importantly, sera were collected earlier after the first immunization (days 0, 6, 22, 34 and 57) (S2 Fig). When the rNS1-based ELISA test was performed, comparable results to sera from TBEV, WNV, and ZIKV immunized mice were obtained for IgM detection, with P/N ratios above the cut-off obtained in all collected sera samples. However, due to the short period of time between the first immunization and the first sample collection (5 days), IgG P/N ratios significantly above the cut-off were only obtained after the second bleeding (day 22) (S3 Fig).

The specificity of each rNS1 was tested against immunized sera (bleeding II) from the other antigens. As an example, IgM from TBEV NS1 immunized sera could be detected up to a 1:1000 dilution (Fig 2E), while IgG up to 1:10.000 (Fig 2F). However, reactivity below cut-off was observed for the rNS1 antigens of WNV, ZIKV, and DENV1-4 (Fig 2E and 2F) demonstrating high specificity towards TBEV rNS1. This analysis was extended to all rNS1 tested and results are represented in the heatmap of Fig 2G and 2H. IgM antibodies in sera samples from immunized mice with WNV-NS1 and ZIKV-NS1 (including TBEV-NS1 explained above) recognized the homologous rNS1 antigen with very low cross-reactivity to a second heterologous antigen (Fig 2G). Comparable results but with stronger reactivity to the homologous rNS1 antigen were observed for IgG detection using the same sera samples (Fig 2H).

Detection of IgM/IgG antibodies using sera samples of immunized mice (bleeding II) with DENV1-4 NS1 to the homologous DENV rNS1 proteins was also detected with very low cross-reactivity to heterologous TBEV, WNV, and ZIKV rNS1 antigens. Some cross-reactivity of IgM/IgG antibodies was observed to rNS1 proteins of DENV serotypes as follow: IgM/IgG antibodies from DENV1 NS1 sera also recognized a heterologous rNS1 protein of DENV-2 and DENV-3. IgM antibodies from DENV2 NS1 sera cross-reacted with the rNS1 protein of DENV-1 and DENV-3 at 1:100 serum, but lower cross-reactivity was observed in the following serum dilutions. IgG antibodies from the same serum sample cross-reacted mainly with the rNS1 protein of DENV-1. IgM and IgG antibodies from DENV-3 NS1 sera mainly cross-reacted with the rNS1 protein of DENV-1. Only IgM/IgG antibodies from DENV-4 NS1 sera showed low cross-reactivity to heterologous rNS1 antigens from other DENV serotypes (Fig 2G and 2H).

These results conducted in immunized mice demonstrate the high sensitivity and specificity of rNS1-based ELISA showing promise for a diagnostic assay in humans.

## Detection of IgM/IgG antibodies from a cohort of patients infected by TBEV

To assess the sensitivity and specificity of the rNS1-based ELISA test for the detection of IgM/IgG antibodies in infected humans, a total of 100 sera samples from RT-PCR confirmed TBEV infected patients were selected. 11 samples for each group were collected during the first and second phase of the infection (1–8 and 9–19 days after onset of symptoms, respectively), 34 sera samples were collected during the acute phase (from 20 days after onset of symptoms to 2 months), and the last 44 samples were obtained from patients in the convalescent phase (from

2 months after onset of the symptoms). All sera samples were tested for the detection of IgM/IgG antibodies by commercial ELISA (plates coated with inactivated TBEV E antigen, Enzygnost; Simens GmbH) and rNS1-based ELISA (plates coated with purified TBEV rNS1 antigens) assays (S2 Table).

Among the 100 samples analyzed by commercial ELISA, 11 samples from the first phase of infection were IgM/IgG negative, while all the samples from the second, acute and convalescent phases were IgM/IgG positive (Fig 3A and 3B, and S2 Table). Comparable results were also obtained by rNS1-based ELISA. 10 samples from the first phase of infection were IgM and IgG negative. Only one sample from the first phase turned IgG positive with low P/N ratio. All the samples from the second phase were IgM/IgG positive. 33/34 and 37/44 samples from acute and convalescent phases, respectively, were IgM positive. All the samples from these two phases were also IgG positive (Fig 3C and 3D, and S2 Table). Cut-off values for commercial ELISA were calculated according to the manufacturer's instructions (IgM cut-off at 0.25 $OD_{450}$ and IgG at 4.1 U/mL). Calculation of the cut-off values for TBEV rNS1-based ELISA was calculated based on the comparative receiver operating characteristic (ROC) curve analysis (IgM cutoff at 2.0 and IgG at 1.4 P/N ratios).

Based on 89 IgM/IgG positive samples (100% sensitivity) and 11 IgM/IgG negative samples (100% specificity) analyzed by commercial ELISA, we determined the sensitivity and specificity of the TBEV rNS1-based ELISA with 95% confidence intervals (CI). The sensitivity was 91% and 100%, while the specificity was 100% and 91% for IgM (95% CI, 0.77 to 0.89) and IgG (95% CI, 0.62 to 0.81) detection, respectively (Fig 3E and 3F). Combined IgM/IgG sensitivity and specificity was 96% among the 100 TBEV-positive specimens tested.

## rNS1-based ELISA for WNV, ZIKV and DENV 1–4 infections

A cohort of 84 RT-PCR-positive specimens from infected patients with flavivirus (travelers firstly exposed to DENV infection) were selected for this study. Among the specimens, 16 samples were positive for WNV, 15 for either ZIKV or DENV1, 2 or 3, and 8 for DENV4. Each sample was tested for the presence of IgM/IgG antibodies by commercial ELISA and rNS1-based ELISA (plates coated with specific purified rNS1 antigens) assays (Fig 4A and 4B, and S3 Table). For rNS1-based ELISA, the positivity of a P/N ratio was determined according to the ROC cut-off value determined for each antigen on 13 negative sera from endemic regions (S1 Table). The optimal cut-off value for IgM detection of WNV and ZIKV fell at 1.4, while for IgG detection it fell at 1.7 and 1.8 P/N ratios, respectively. For the detection of IgM and IgG antibodies against all DENV serotypes, both cut-off values fell at 1.3 P/N ratio.

12/16 IgM and 9/16 IgG positive WNV sera samples were diagnosed by a commercial ELISA (plates coated with recombinant WNV E antigen, Focus Diagnostics). Whereas, 10/16 IgM and 12/16 IgG positives were obtained by WNV rNS1-based ELISA. 11/15 IgM and 14/15 IgG ZIKV positive sera samples were detected by a commercial ELISA (plates coated with ZIKV recombinant NS1 antigen, Euroimmune; Labordiagnostika AG), while all fifteen ZIKV sera samples were IgM/IgG positive when they were tested by the ZIKV rNS1-based ELISA. Specific DENV serotypes samples were analyzed for the presence of IgM/IgG antibodies using the same commercial ELISA assay (plates coated with DENV type 2 E antigen, NovaTec, immunodiagnostic GmbH). 10/15 IgM and 15/15 IgG for DENV1, 11/15 IgM and 15/15 IgG for DENV2 and 10/15 IgM and 13/15 IgG for DENV3 resulted positive, while 3/8 IgM and 8/8 IgG resulted positive for DENV4. In comparison, using the rNS1-based ELISA (plates coated with DENV serotype-specific rNS1 antigen), all 15 sera samples from DENV1 and DENV2, and 8 sera from DENV4 infected patients were positive for IgM detection, and 13/15, 14/15,

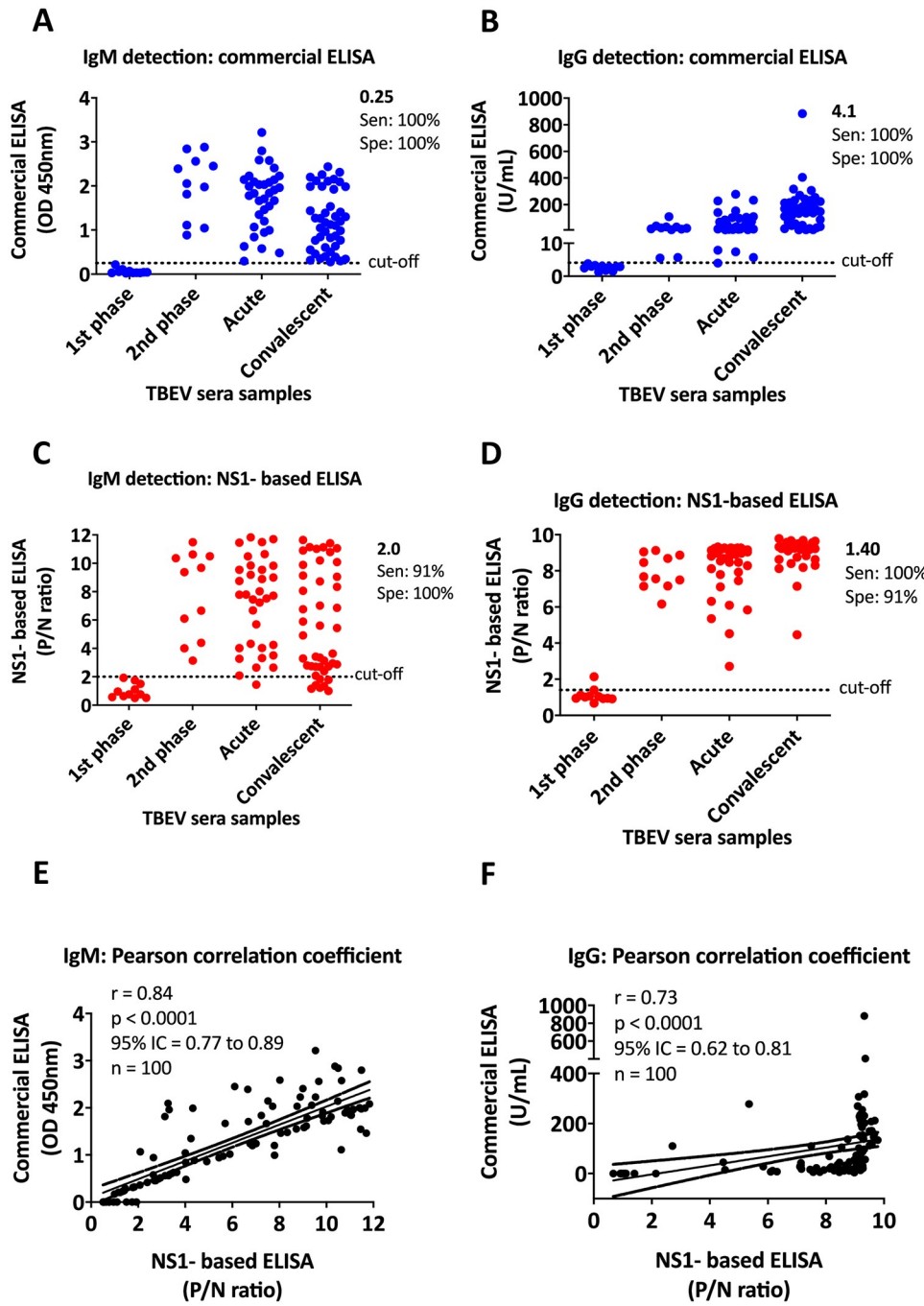

**Fig 3. Detection of IgM/IgG antibodies from TBEV infected individuals.** A and B) Detection of IgM (A) IgG (B) antibodies by commercial ELISA. IgM results are reported as $OD_{450}$, while IgG results are reported as U/mL. Cut-off values for IgM (0.25) and IgG (4.1) were calculated according to the manufacturer's instructions. C and D) Detection of IgM (C) IgG (D) antibodies by TBEV rNS1-based ELISA. Optimal cut-off values of the P/N ratio ($OD_{450}$ of test specimen divided by the mean $OD_{450}$ of negative control specimens) were calculated based on the comparative receiver operating characteristic (ROC) curve analysis. Cut-off values for IgM and IgG fell at 2.0 and 1.40, respectively. E and F) Correlation between commercial and TBEV rNS1-based ELISA assays. The two-tailed Pearson's correlation value (r) was calculated for IgM and IgG values. A *P* value of <0.0001 rejected the null hypothesis that there exists no correlation between commercial and rNS1-based ELISA methods. 95% Interval Confidence (IC) value is also indicated and showed in dotted lines above and under the linear correlation. 100 sera samples (n) from different phases of TBEV infection were included in the analysis. * each ELISA result includes the average of two biological replicates.

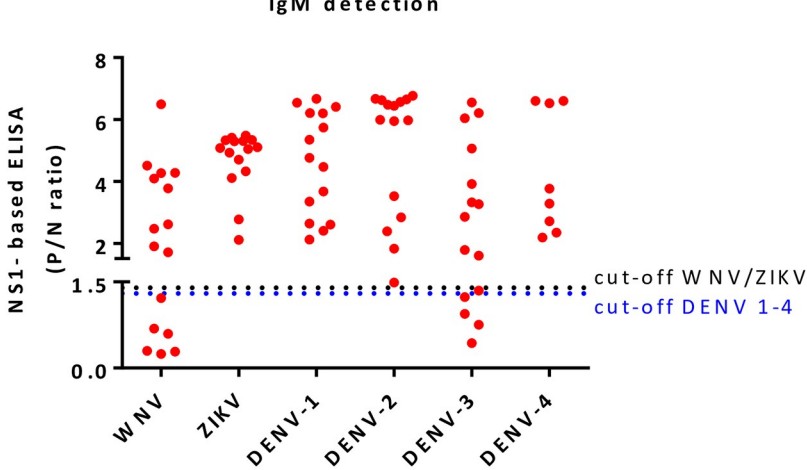

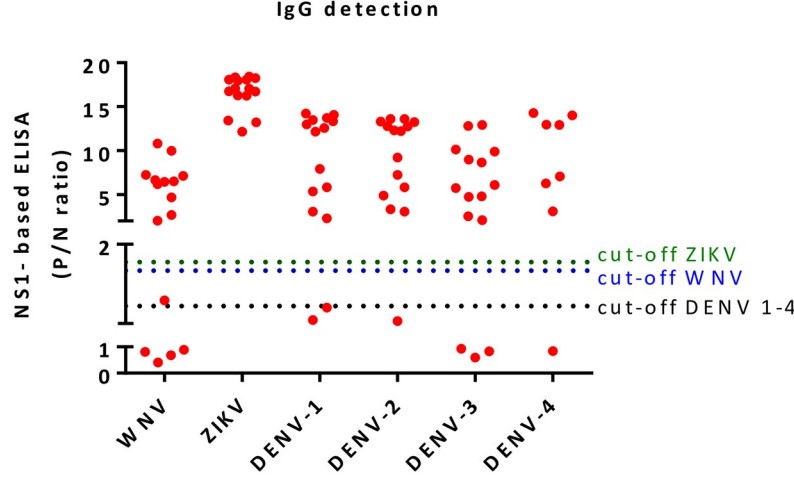

**Fig 4. Detection of IgM/IgG antibodies from WNV, ZIKV and DENV 1–4 infected individuals.** A and B) Detection of IgM (A) IgG (B) antibodies by rNS1-based ELISA. Each group of RT-PCR confirmed patients were screened for the presence of IgM/IgG antibodies by rNS1-based ELISA tests using specific purified rNS1 antigens. Optimal cut-off values of P/N ratios were calculated based on ROC curve analysis. Each cut-off was selected based on the P/N ratio value, which gave 100% sensitivity and specificity. * each ELISA result includes the average of two biological replicates.

and 7/15 were IgG positive, respectively. Among 8 samples from DENV4 infected patients, only one sample was IgG negative, while all the others were IgM and IgG positive.

## Differential serodiagnosis of TBEV by rNS1-based ELISA

43 human sera samples were tested for the presence of IgM/IgG antibodies by commercial ELISA (plates, coated with inactivated TBEV E antigen, Enzygnost; Simens GmbH) and TBEV

rNS1-based ELISA assays (Fig 5A and 5B, and S4 Table). Of the 43 specimens, 10 healthy and 3 sera samples of individuals from endemic areas who got the YFV vaccine (live attenuated 17D vaccine) were included as controls. All 13 sera samples were from TBEV non-endemic regions. The other 30 sera samples include 10 individuals who got the TBEV vaccine (a suspension of purified TBE inactivated virus), 10 patients with TBEV acute-phase infection and 10 sera from vaccine breakthrough (VBT) patients.

All samples from YFV and TBEV vaccinated groups were IgM negative, while 10/10 and 7/10 sera samples from TBEV acute-phase infection or VBT were IgM positive by commercial ELISA, respectively. Detection of IgG antibodies by the commercial ELISA test showed all sera samples were IgG positive, including 3 sera samples from the YFV vaccinated group (S4 Table). But, only sera samples from TBEV acute-phase infection and VBT patients were all IgM and IgG positive when they were analyzed by TBEV rNS1-based ELISA (Fig 5A and 5B, and S4 Table).

## Discussion

Cross-reactivity between different flaviviruses has been the main drawback in the serological assays available for the diagnosis of flavivirus infections [10, 28]. Many of them are based on the detection of antibodies against envelope protein that poorly allow the differentiation among flavivirus infections [29]. A lot of effort has been dedicated generating recombinant envelope proteins or virus-like particles containing mutations in the fusion loop domain to reduce the detection of cross-reacting antibodies [30–33]; however the problem still remains unsolved. Evidence has been provided that the use of rNS1 proteins for the detection of IgM/IgG antibodies is highly specific, as demonstrated using commercial kits or NS1 proteins obtained commercially [12–14]. However, contradictory results in the detection of IgM antibodies from patients of flavivirus-endemic regions keep in debate the usefulness of NS1 as antigen for antibody detection [34]. In this work, we compared homogeneously purified rNS1 antigens across representative members of the family for sensitivity and specificity to determine the usefulness of the approach in flavivirus serology.

Different strategies for the production and purification of flavivirus rNS1 proteins and their use in the development of NS1-based ELISA assays have been reported, based on rNS1 proteins produced in systems that are prone to problems of protein stability and lack the proper folding and post-translational modifications [35–37]. In the present study, we report a strategy for the production and purification of rNS1 proteins of TBEV, WNV, ZIKV and all DENV serotypes from the supernatant of transiently transfected HEK293T cells. Recombinant proteins present in the supernatant were purified by affinity chromatography. SDS-PAGE and western blot analysis of purified proteins showed a high oligomeric status and glycosylation profile comparable to NS1 protein secreted from infected cells [17, 21]. Purified rNS1 proteins were validated for the detection of specific IgM/IgG antibodies using sera from immunized mice. Results showed high specificity of rNS1 antigens for the detection of IgM/IgG antibodies with TBEV, WNV, and ZIKV. Comparable results but with cross-reactivity to a second DENV heterologous rNS1 antigen were obtained using sera from immunized mice with DENV1-4, most likely due to the high degrees of sequence identity between DENV serotypes [38]. rNS1 proteins were also tested for the detection of IgM/IgG antibodies using well-characterized human sera. 100 sera collected from patients with TBE at different phases after the onset of symptoms revealed a combined sensitivity and specificity (IgM/IgG) of 96% by TBEV rNS1-based ELISA. Although, similar results were obtained in all the samples from each phase analyzed by commercial ELISA and TBEV NS1-based ELISA assays, 7 samples from the convalescent phase of TBEV infection were IgM negative while they were considered positive by commercial assays, presumably because all these samples were collected from two months

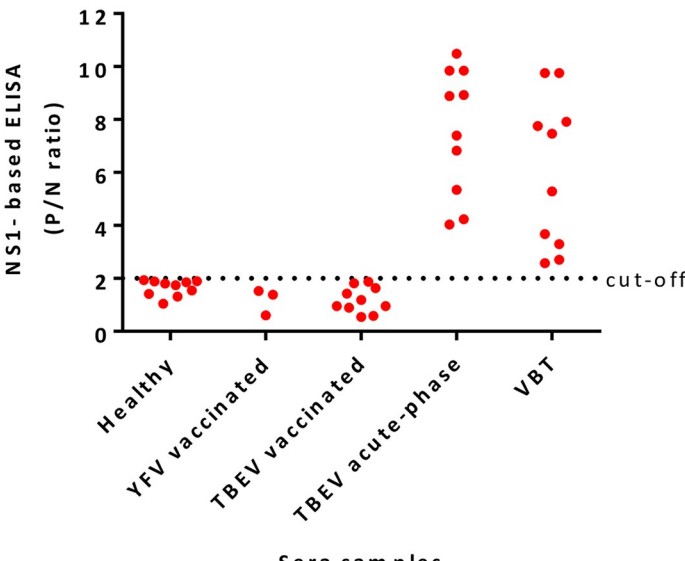

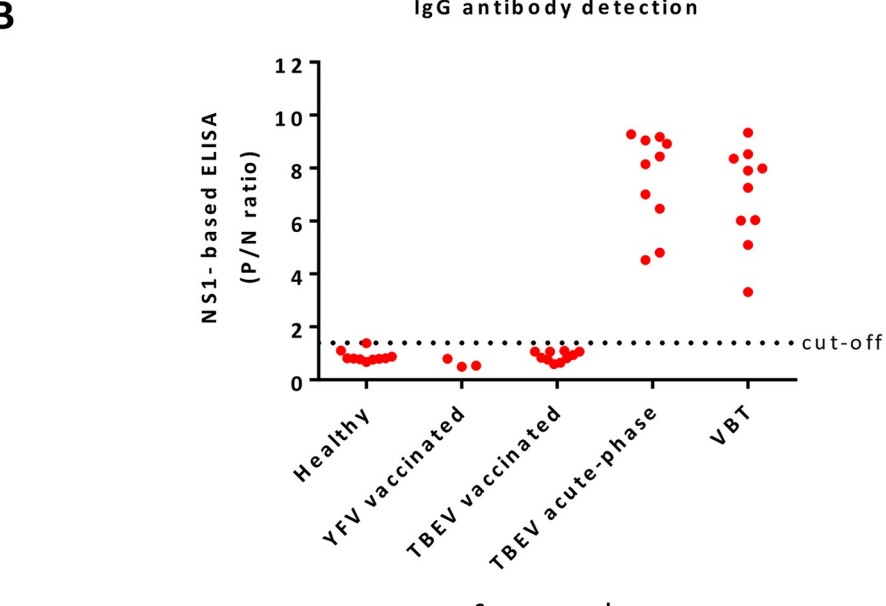

**Fig 5. Differential serological diagnosis of TBEV.** A and B) Detection of IgM (A) IgG (B) antibodies by TBEV rNS1-based ELISA using sera samples from five different groups of sera samples: 10 sera samples of healthy individuals from non-endemic TBEV regions and 3 sera from YFV vaccinated individuals (TBEV non-endemic areas) were used as controls. TBEV vaccinated group corresponds to 10 sera from individuals who got the vaccine (a suspension of purified TBE inactivated virus). The acute group corresponds to individuals with acute TBEV infection. The vaccine breakthrough (VBT) group corresponds to individuals who got the vaccine and were also infected with TBEV. * each ELISA result includes the average of two biological replicates.

after onset of symptoms, i.e. when IgM antibodies decrease to undetectable levels. Unlike results obtained from commercial ELISA, only one serum from the first phase of infection was IgG positive by TBEV rNS1-based ELISA, albeit at a low titer.

The extensive comparison by testing TBEV-infected sera samples from first, second, acute and convalescent phases simultaneously with commercial ELISA and TBEV rNS1-based ELISA assays showed a high degree of correlation between both assays for the detection of IgM antibodies. However, a low correlation was obtained for the detection of IgG antibodies. Although 89/89 IgG positive and 11/11 IgG negative TBEV-confirmed sera specimens from first, second, acute and convalescent phases showed high or low P/N ratio values, respectively, by TBEV rNS1-based ELISA, variable values by commercial ELISA were obtained (range, 5.7 to 883 U/mL). High variability between commercial ELISA assays has been already reported [39, 40]; however, the performance of commercial ELISA and TBEV rNS1-based ELISA assays in terms of a total number of positives or negative samples was comparable.

rNS1-based ELISA for virus-specific IgM/IgG detection performed well for WNV, ZIKV and all DENV serotypes. With the exception of the Eurimmune ZIKV assay based on NS1 protein, all the other commercial ELISA assays are based on envelope proteins. In almost all cases, low sensitivity for IgM detection was observed as has been already demonstrated for several commercial assays [40, 41]. Lower sensitivity in particular for DENV4 could be explained by the fact that most of the commercial assays for DENV IgM detection are based only on the envelope protein of DENV2, one of the most spread DENV serotypes [42]. This was not the case when samples were analyzed by DENV4 rNS1-based ELISA since all DENV4 sera were also IgM positive for DENV4.

Differential diagnosis between vaccinated and naturally infected patients with TBEV is very important to discriminate vaccine breakthrough cases. However, the commercial ELISA assays available for the diagnosis of TBEV are based on structural proteins, an antigen that is also present in TBEV vaccines and is highly cross-reactive between flaviviruses [29, 43]. Cross-reactivity was observed in 3 sera samples from individuals vaccinated against YFV. These samples were tested IgG positive by commercial ELISA; however, they resulted negative by TBEV rNS1-based ELISA. We took advantage of the rNS1-based ELISA for the specific detection of IgM/IgG antibodies against TBEV-rNS1 protein from sera samples of TBEV vaccinated and vaccine breakthrough (VBT) patients, a group of vaccinated people above the age of 50 years that got infected with TBEV after vaccine failure. As demonstrated in the present study, all TBEV vaccinated samples were IgG positive by commercial assays, while they were negative by TBEV rNS1-based ELISA. When VBT sera were tested with commercial ELISA, they were characterized by high IgG response with low IgM levels (in many cases undetectable) against envelope protein. Interestingly, detection of IgM/IgG antibodies from this group of samples against the rNS1 protein of TBEV showed higher sensitivity compared to commercial ELISA. Together these results indicate the usefulness of the TBEV rNS1-based ELISA for differential diagnosis of TBEV.

In conclusion, our study showed that the rNS1-based ELISA is a sensitive and highly specific test for the detection of IgM/IgG antibodies indicating its potential use in serodiagnosis and surveillance studies. The rNS1-based ELISA could be also transferred to a point of care test for syndrome-based multi-parametric diagnosis of flavivirus, an urgent test especially in those regions where more than one flavivirus is circulating [44].

## Materials and methods

### Cloning, expression, and purification of rNS1

The nucleotide sequences coding for NS1 proteins were derived from the NCBI database entry of TBEV Neudoerfl strain (NC_001672.1), WNV NY99 strain (NC_009942.1), ZIKV SPH2015

strain (KU321639.1), DENV1 Hawaii A strain (NC_001477.1), DENV2 New Guinea C strain (NC_001474.2), DENV3 Sri Lanka strain (NC_001475.2) and DENV4 TC25 strain (NC_002640.1). Sequences were codon-optimized for the expression in *Mus musculus* and were synthesized commercially by *Gene Art Gene Synthesis* (Thermo Fisher Scientific). Only for WNV-NS1 two-point mutations were inserted (RQ10NK), which have been shown to increase secretion [25]. Codon-optimized sequences of NS1 were purchased as synthetic genes in pMA-T vectors (Thermo Fisher Scientific) and sub-cloned into the pcDNA3.1 expression vector (Life Technologies) fused to an immunoglobulin leader sequence (Sec) at the N-terminus. The genes were kept in-frame with the polyhistidine (6x-His) tag or the V5 tag (GKPIPNPLLGLD) at the C-terminus.

Transient transfection of HEK293T cells was performed by the standard calcium phosphate method [45]. Briefly, freshly replated low-passage cells were overalaid with the calcium-phosphate-DNA suspension prepared by mixing plasmid DNA (2.5μg and 20 μg for 6 wells plates and 150mm plates, respectively) with 2.5M NaCl and 2X Hepes buffered saline buffer (HBS pH 7.1). 16h after transfection, cells were washed twice in phosphate-buffered saline (PBS) and further cultured for 30h in serum-free media supplemented with 5 mM of sodium butyrate. Culture supernatants were then cleared by centrifugation at 4000 g for 4 min, while total cellular extracts were lysed in TNN buffer (100 mM Tris-HCl, pH 8, 250 mM NaCl, 1% NP-40) supplemented with Protease Inhibitor Cocktail (PIC, Sigma, P8340) at 4˚C. Cellular extracts were kept at -20˚C until use, meanwhile culture supernatant was used immediately for protein purification.

Purification of recombinant NS1 (rNS1) from clarified culture supernatants was performed by affinity chromatography on FPLC using HiTrap Chelating HP 5mL columns (GE Healthcare). Eluted rNS1 in Imidazole were concentrated and buffer-exchanged to PBS using Ultra-4 centrifugal filters devices (Amicon, 10K).

Analysis of the protein was performed by sodium dodecyl sulfate polyacrylamide gel electrophoresis (SDS-PAGE). Cell lysates/supernatant were boiled at 95˚C for 12 min (or left unheated when the samples were run under non-denaturing or native conditions), and centrifuged for 1 min at RT at 1000g before loading the 10% SDS PAGE. Gels were run in SDS electrophoresis buffer (25 mM Tris, 190 mM glycine, 0.1% SDS), initially at 90 V into the stacking gel and later at 130 V into the running gel. Native PAGE was prepared without SDS neither in the gel, nor in the loading and running buffers. 2-Mercaptoethanol in the loading buffer was only present when the samples were analyzed under normal conditions.

The purity and concentration of the purified proteins were estimated by Coomassie blue staining and by Bradford assay, respectively. Densitometric analysis was performed using Image Lab™ Software 6.0.1 (Bio-Rad).

Immunoblots were performed using monoclonal antibodies anti-6x-His-tag (Sigma) or V5 tag (Provided by Dr. Oscar Burrone). As loading control rabbit antibody anti-actin (Sigma) was used. Nitrocellulose membrane (GE Healthcare—10600015) was blocked for 1 hour in 5% milk followed by incubation with the anti Histidine/V5 tag primary antibodies diluted in 5% milk/0,5% Tween-20 for 1h at room temperature. After three washes with TBS 0.5% Tween-20, secondary antibodies conjugated with HRP were diluted in 5% milk/0,5% Tween-20 and incubated for 1 hour at room temperature. Blots were developed using Immobilon Western Chemiluminescent HRP Substrate according to manufacturers' instructions. Signals were visualized by ECL (ThermoFisher-Pierce, Rockford, IL, USA).

## Glycosylation analysis

Purified NS1 proteins were digested for 1.5 hours with endoglycosidase Hf (Endo Hf) or/and Peptide-N-Glycosidase-F (PNGase) endoglycosidases according to manufacturer's protocols

(New England Biolabs). Reactants were then subjected to 10% SDS-PAGE under reducing conditions followed by western blot with anti-His mAb.

## Mice immunization

Four 5–6 weeks old, female Balb/c mice per each condition were immunized intradermally with a plasmid encoding for rNS1 by Gene Gun technology (Bio-Rad, Hercules, CA, USA). A scheme based on prime-boost immunization was followed. Before the immunization, the abdominal area of each mouse was shaved and 1 μm gold particles coated with 1 μg of plasmid DNA (V5-tagged constructs of NS1 TBEV, WNV, ZIKV, and all DENV serotypes, 1–4) were delivered by biolistic particle system at 400 psi. Each group of mice was immunized 4 times with a specific plasmid at fourteen days intervals. Blood samples were collected by sub-mandibular puncture before immunization (pre-immune sera) and 5–7 days after each boost (bleeding I, II, III and IV). Sera samples were collected and stored at -20˚C until use.

## Clinical samples

Sera samples of TBEV and WNV infected patients were anonymously obtained from endemic areas in Slovenia and Italy, respectively. While the ZIKV and DENV1-4 positive sera samples originate from Slovenian travelers. ZIKV confirmation and serotyping of DENV patients was based on RT-PCR detection. The reference test for the TBEV sera samples (n = 100) included Enzygnost Anti-TBE/FSME ELISA. Results are summarized in S2 Table. As seen, for 11 sera samples from the first phase of TBEV infection the ELISA was negative. These samples were used to calculate the specificity of the rNS1-based ELISA, while 89 IgM/IgG positive sera samples were used to calculate the sensitivity. WNV-positive sera samples (n = 16) were tested for the presence of IgM/IgG antibodies with the Focus Diagnostics ELISA. The ZIKV-positive samples (n = 15) were tested with the Euroimmune ELISA, and DENV sera samples (n = 53, 15 samples for either serotype 1, 2 or 3 and 8 samples for serotype 4) were tested using Nova-Tec ELISA.

## Ethics statement

Human sera were obtained from an established collection at the Institute of Microbiology and Immunology of the Faculty of Medicine at the University of Ljubljana, Slovenia, that was approved by the Medical Ethics Committee of the Ministry of Health of the Republic 1 of Slovenia (No. 0120-188/2018/6 and No. 152/06/13).

Animal care and treatment were conducted in conformity with institutional guidelines after approval by the ICGEB Institutional Review Board (project number AWB2015/1) following consent from the Italian Ministry of Health in accordance with the Italian law (D.lgs. 26/2014), following European Union policies (European and Economic Council Directive 86/609, OJL 358, December 12, 1987).

## NS1-based enzyme-linked immunosorbent assay (rNS1-based ELISA)

Nunc Maxi Sorp Immuno-Plates (ThermoFisher-Nunc, Roskilde, Denmark) were coated with 5 μg/ml of purified rNS1 6xHis-tagged antigens in 50 mM $Na_2CO_3$/$NaHCO_3$ buffer pH 9.6 and incubated overnight at 4˚C. Plates were washed with Phosphate-Buffered Saline (PBS) buffer and non-specific binding sites were blocked with 2% milk in PBS for 45 min at room temperature (RT). After washing, 100 μl of 1:100 sera dilutions (or serial dilutions when it is indicated) of sera from immunized mice or 1:20 dilutions of sera from infected humans were added to each well. For IgM detection, sera samples were incubated for 1 hour at 37˚C, while

for IgG detection sera samples were incubated for 1 hour at RT. 100 μl/well of HRP-linked goat antibodies anti-mouse IgM/IgG or anti-human IgM/IgG were used (Sigma, 1:5000). In all cases, secondary antibodies were incubated for 1 hour at RT. After each antibody incubation, wells were washed three times with PBS 0.1% tween 20 (PBST). Signal was developed by adding 70 μl of 3,3',5,5'-Tetramethylbenzidine (TMB) substrate (Sigma). The reaction was stopped by adding 30 μl of 2N $H_2SO_4$ to each well. The optical density was measured at 450 nm ($OD_{450}$) with an ELISA EnVision 2104 Multilabel Plate Reader (Perkin Elmer). The P/N ratio for both IgM/IgG rNS1-based ELISA was obtained by dividing the $OD_{450}$ of test specimen divided by the mean $OD_{450}$ of 16 negative control specimens.

### Statistical analysis

All statistical analysis and graphs were performed and generated using GraphPad Prism software. The comparative receiver operating characteristic (ROC) curve analysis was used to calculate the optimal cut-off values of P/N ratio ($OD_{450}$ of test specimen divided by the mean $OD_{450}$ of negative control specimens) for IgM/IgG detection by rNS1-based ELISA. A bin range of different P/N ratio values was used to select the optimal cut-off value that gave 100% sensitivity and specificity.

### Supporting information

**S1 Fig. rNS1-based ELISA test for the detection of IgM/IgG antibodies against WNV/ ZIKV rNS1 6xHis-tagged proteins.** A and B) Detection of IgM/IgG antibodies from 4 different immunized mice with WNV NS1 V5-tagged construct. C and D) Detection of IgM/IgG antibodies from 3 different immunized mice with ZIKV NS1 V5-tagged construct. Plates were coated with purified WNV-rNS1 and ZIKV-rNS1 antigens for the detection of antibodies from mice immunized with WNV-NS1 and ZIKV-NS1, respectively. * each ELISA result includes the average of two biological replicates.
(TIF)

**S2 Fig. Schematic representation of the protocol followed for mice immunization with DENV constructs.** 4 different BALB/c mice for each DENV serotype (DENV-1, DENV-2, DENV-3, and DENV-4) were immunized with specific NS1 V5-tagged plasmids by gene gun technology. The day when the prime-boost immunization was performed, and sera samples were collected for IgM/IgG detection are indicated in numbers.
(TIF)

**S3 Fig. rNS1-based ELISA test for the detection of IgM/IgG antibodies against DENV1-4 rNS1 6xHis-tagged proteins.** A to H) Detection of IgM/IgG antibodies from 4 different immunized mice with NS1 V5-tagged construct for each DENV serotype (DENV-1, DENV-2, DENV-3, and DENV-4). Plates were coated with purified DENV1-rNS1, DENV2-rNS1, DENV3-rNS1 and DENV4-rNS1 antigens for the detection of antibodies from mice immunized with each DENV serotype, respectively. * each ELISA result includes the average of two biological replicates.
(TIF)

**S1 Table. Optimal cut-off values of P/N ratio for IgM/IgG antibody detection from sera of immunized mice and infected patients.** P/N: positive to the negative ratio ($OD_{450}$ of test specimen divided by the mean $OD_{450}$ of negative control specimens). ROC: comparative receiver operating characteristic curve rNS1: recombinant non-structural protein 1 TBEV: Tick-borne

encephalitis virus WNV: West Nile virus ZIKV: Zika virus DENV1-4: Dengue virus serotype 1, serotype 2, serotype 3 and serotype 4.
(TIF)

**S2 Table. Number of positive samples from TBEV infected patients.** TBEV: Tick-borne encephalitis virus ELISA: enzyme-linked immunosorbent assay rNS1: recombinant non-structural protein 1 [a] A total of 11 samples were tested for each group, first (1st) and second (2nd) phase of TBEV infection [b] A total of 34 samples were tested for the acute phase of TBEV infection [c] A total of 44 samples were tested for the convalescent phase of TBEV infection [d] plates were coated with inactivated TBEV E antigen, Enzygnost; Simens GmbH [e] plates were coated with recombinant non-structural protein 1 of TBEV.
(TIF)

**S3 Table. Number of positive samples from flavivirus infected patients.** WNV: West Nile virus ZIKV: Zika virus DENV1-4: Dengue virus serotype 1, serotype 2, serotype 3 and serotype 4 ELISA: enzyme-linked immunosorbent assay rNS1: recombinant non-structural protein 1 [a] IgM/IgG antibodies were detected by using commercial kits according to the manufacturer's instructions for WNV (plates coated with recombinant WNV E antigen, Focus Diagnostics), ZIKV (plates coated with ZIKV recombinant NS1 antigen, Euroimmune; Labordiagnostika AG) and, DENV1-4 (plates coated with DENV type 2 E antigen, NovaTec, immunodiagnostic GmbH). [b] IgM/IgG antibodies were detected using purified rNS1 proteins of WNV, ZIKV and all DENV serotypes.
(TIF)

**S4 Table. Number of TBEV positive IgM/IgG samples out of the total number of samples analyzed by commercial ELISA and rNS1-based ELISA assays.** TBEV: Tick-borne encephalitis virus ELISA: enzyme-linked immunosorbent assay rNS1: recombinant non-structural protein 1 VBT: vaccine breakthrough NA: non-available [a] IgM/IgG antibodies were detected by using a commercial kit according to the manufacturer's instructions for TBEV (plates were coated with inactivated TBEV E antigen, Enzygnost; Simens GmbH [b] IgM/IgG antibodies were detected by rNS1-based ELISA coating the plates with the rNS1 protein of TBEV.
(TIF)

## Acknowledgments

We are thankful to Stefano Artico, Monica Poggianella and Mateja Jelovšek for highly professional technical assistance.

## Author Contributions

**Conceptualization:** Alessandro Marcello.

**Data curation:** Erick Mora-Cárdenas, Pierlanfranco D'Agaro, Tatjana Avšič-Županc, Alessandro Marcello.

**Formal analysis:** Alessandro Marcello.

**Funding acquisition:** Tatjana Avšič-Županc, Alessandro Marcello.

**Investigation:** Erick Mora-Cárdenas, Chiara Aloise, Valentina Faoro, Nataša Knap Gašper, Miša Korva, Ilaria Caracciolo, Alessandro Marcello.

**Methodology:** Erick Mora-Cárdenas, Alessandro Marcello.

**Project administration:** Alessandro Marcello.

**Resources:** Pierlanfranco D'Agaro, Tatjana Avšič-Županc, Alessandro Marcello.

**Supervision:** Alessandro Marcello.

**Writing – original draft:** Erick Mora-Cárdenas, Alessandro Marcello.

**Writing – review & editing:** Alessandro Marcello.

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
