## [Decision Letter · Decision Letter 0]

18 Oct 2019

Dear Dr. Marcello:

Thank you very much for submitting your manuscript "Comparative specificity and sensitivity of NS1-based serological assays for the detection of Flavivirus immune response" (PNTD-D-19-01247) for review by PLOS Neglected Tropical Diseases. Your manuscript was fully evaluated at the editorial level and by independent peer reviewers. The reviewers appreciated the attention to an important topic but identified some aspects of the manuscript that should be improved.

We therefore ask you to modify the manuscript according to the review recommendations before we can consider your manuscript for acceptance. Your revisions should address the specific points made by each reviewer.

(1) A letter containing a detailed list of your responses to the review comments and a description of the changes you have made in the manuscript.

(2) Two versions of the manuscript: one with either highlights or tracked changes denoting where the text has been changed (uploaded as a "Revised Article with Changes Highlighted" file ); the other a clean version (uploaded as the article file).

(3) If available, a striking still image (a new image if one is available or an existing one from within your manuscript). If your manuscript is accepted for publication, this image may be featured on our website. Images should ideally be high resolution, eye-catching, single panel images; where one is available, please use 'add file' at the time of resubmission and select 'striking image' as the file type. 

Please provide a short caption, including credits, uploaded as a separate "Other" file. If your image is from someone other than yourself, please ensure that the artist has read and agreed to the terms and conditions of the Creative Commons Attribution License at http://journals.plos.org/plosntds/s/content-license (NOTE: we cannot publish copyrighted images). 

(4) Appropriate Figure Files 

Please remove all name and figure # text from your figure files upon submitting your revision. Please also take this time to check that your figures are of high resolution, which will improve both the editorial review process and help expedite your manuscript's publication should it be accepted. Please note that figures must have been originally created at 300dpi or higher. Do not manually increase the resolution of your files. For instructions on how to properly obtain high quality images, please review our Figure Guidelines, with examples at: http://journals.plos.org/plosntds/s/figures

While revising your submission, please upload your figure files to the Preflight Analysis and Conversion Engine (PACE) digital diagnostic tool, https://pacev2.apexcovantage.com/ PACE helps ensure that figures meet PLOS requirements. To use PACE, you must first register as a user. Then, login and navigate to the UPLOAD tab, where you will find detailed instructions on how to use the tool. If you encounter any issues or have any questions when using PACE, please email us at figures@plos.org.

We hope to receive your revised manuscript by Dec 17 2019 11:59PM. If you anticipate any delay in its return, we ask that you let us know the expected resubmission date by replying to this email.

To submit your revised files, please log in to https://www.editorialmanager.com/pntd/

Sincerely,

Sassan Asgari

Guest Editor

A. Desiree LaBeaud

Deputy Editor

Reviewer's Responses to Questions

**Key Review Criteria Required for Acceptance?**

**Methods**

-Are the objectives of the study clearly articulated with a clear testable hypothesis stated?

-Is the study design appropriate to address the stated objectives?

-Is the population clearly described and appropriate for the hypothesis being tested?

-Is the sample size sufficient to ensure adequate power to address the hypothesis being tested?

-Were correct statistical analysis used to support conclusions?

-Are there concerns about ethical or regulatory requirements being met?

Reviewer #1: Objectives and study design is very well described. Statistical analysis was appropriate. Human and animal ethic approvals are provided.

Reviewer #2: The study by Mora-Cardenas et al, evaluates the utility of antibodies to distinct flavi rNS1 proteins to be used detect seropositive response to distinct flavi viruses. The work is not especially novel as the use of NS1 specific antibodies to detect and identify exposure to different flavi viruses have been reported by multiple groups previously. In this study the authors specifically evaluated recombinant NS1 expressed in 293HEK cells from several flaviviruses including tick-borne encephalitis, west nile, zika and dengue virus (1-4). However it was somewhat surprising that JE and YF were not included. The initial analysis examined the purified proteins. It was well performed and demonstrated the successful expression of the mature glycosylated hexamer form in the culture supernatan

Reviewer #3: Mora-Cardenas et al have clearly articulated the goals of the presented study. I have some concerns over the generally small sample sizes for human samples for dengue Zika and WNV.

**Results**

-Does the analysis presented match the analysis plan?

-Are the results clearly and completely presented?

-Are the figures (Tables, Images) of sufficient quality for clarity?

Reviewer #1: The results are very clearly presented and fulfill the objectives put forward by the researchers

Reviewer #2: There are some minor inconsistencies that will be shared in the comments to the authors. The results are generally clear. the figures could use some additional clarification or annotations.

Reviewer #3: The describe a NS1 expression from HEK293 cells however they fail to provide details of the transfection. This need to be detailed in the methods section. The authors state they have produced “fully antigenic native recombinant NS1 proteins….” Remove native from this statement. Furthermore, in figure 1 where authors perform western blot analysis the proteins are probed with anti-V5 or anti-His antibody not an anti-NS1 antibody. While it is highly likely these antigens are antigenically correct, to confirm this the recombinant NS1 proteins should interrogated with MAbs to each domain. The only data in the paper to suggest these antigens are antigenically similar to their native counter parts is the reactivity to human sera. Size exclusion chromatography should be performed to confirm the hexameric state of the expressed NS1 protein.

Likewise, more detail is required in the explanation of the endo H and PNGase F digestions, ie molecular weight shifts, presence of high mannose carbohydrates on endo H samples, not observed in WNV…. This needs more detail in the figure, the text and caption.

Figure 2 C and D Rx, Lx, RxLx are not described in detail to adequately inform the reader

While not critical to reproducing the results from the manuscript, authors have omitted all details of SDS-PAGE gel analysis in the methods

**Conclusions**

-Are the conclusions supported by the data presented?

-Are the limitations of analysis clearly described?

-Do the authors discuss how these data can be helpful to advance our understanding of the topic under study?

-Is public health relevance addressed?

Reviewer #1: Mora-Cardenas et al present a straightforward study investigating the use of the NS1 protein as a diagnostic antigen for a range of flaviviruses. Although NS1 has been well investigated before, the wide range of different flaviviruses utilized in this work (including both mosquito and tick borne viruses) makes this a unique and valuable addition to the literature. In addition to validation of the antigens in mouse models, the researchers present data from a range of human clinical samples to support the use of recombinant NS1 for both IgM and IgG flavivirus serology.

Reviewer #2: The evaluation of specificity and sensitivity are somewhat limited due to the availability of well characterized human samples. Nonetheless the studies are in agreement with similar studies on th utility of NS1 proteins and the antibodies generated to them post infections to evaluate individuals serostatus to different flaviviruses.

Reviewer #3: The conclusions are generally supported by the data presented with the study. As stated above the small numbers of dengue, Zika and WNV is a weakness. The TBE results support the conclusions. The ability to differential between natural infections, vaccinations by NS1 serology is a clear beneficial outcome to the field.

**Editorial and Data Presentation Modifications?**

Reviewer #1: Abstract – capitalization on West Nile, Zika

Page 2 line 7 – individuals

Author summary. First sentence reword

Page 5 line 5 - insert commas “, from mammalian cells,”

Page 5 line 6 - remove "also"

Page 6 line 19 and Fig 1. "non-denaturing" normally refers to without SDS. Suggest changing to unboiled or unheated to clarify. Could also describe the usual TBEV result as a SDS resistant hexameric species

Page 6 line 30 – change "contain the N-linked glycans" to "are modified with N-linked glycans". The specific glycosylation sites are not confirmed, but the data can show that the proteins are being glycosylated at a whole protein level.

Fig1 – no need for the domain labeling, this is inaccurate and not relevant for this work in any case

Fig1 C – native gel analysis, not possible to confirm the oligomeric species – crosslinking and SDS-PAGE or alternatively size exclusion chromotography would have been better, perhaps worth adding a point to the results/discussion near the unusual TBEV result and possible future investigation/validation.

Reviewer #2: Although this work is not especially novel, the characterization of the rNS1 proteins was well performed and might be of interest in comparison those available commercially and expressed in different systems used in other studies.

Reviewer #3: See results section for suggestion to improve clarity of figure 1 and potential additional experiments for the characterisation of NS1.

**Summary and General Comments**

Reviewer #1: This is a straightforward study describing the use of recombinant NS1 as a flavivirus diagnostic antigen. The recombinant NS1 is validated biochemically and validated using murine immunization and serological analysis. The analysis of a range of different clinical samples with comparisons with commercial kits is a particular strong point of the study. Overall the experimental work well is designed and results and discussion clearly presented.

Reviewer #2: The study by Mora-Cardenas et al, evaluates the utility of antibodies to distinct flavi rNS1 proteins to be used detect seropositive response to distinct flavi viruses. The work is not especially novel as the use of NS1 specific antibodies to detect and identify exposure to different flavi viruses have been reported by multiple groups previously. In this study the authors specifically evaluated recombinant NS1 expressed in 293HEK cells from several flaviviruses including tick-borne encephalitis, west nile, zika and dengue virus (1-4). However it was somewhat surprising that JE and YF were not included. The initial analysis examined the purified proteins. It was well performed and demonstrated the successful expression of the mature glycosylated hexamer form in the culture supernatant. Although NS1 proteins are available commercially from a variety of sources, the conformation of these proteins (monomer, dimer and hexamer) is reported to vary greatly depending on the sources and expression system used. The authors might consider including some of these in the biochemical characterization to elucidate some of the potential differences. 

The immunogenicity assessment of the proteins was evaluated mice following immunization with the expression vector (DNA) rather than the purified protein. It was unclear why this approach was performed as differences in expression could result in varying levels of protein expression. The authors also employed different bleed schedules for the mice receiving the dengue constructs compared to those receiving the TBE, WNE and Zika constructs. Lastly the authors examined the specificity of the rNS1 proteins in ELISA-based methods in comparison to commercial based assays. These assay were limited by the number of available samples and often focused on samples from individuals with confirmed exposure to that flavivirus and those from flavi naïve individuals. In these analyses the sensitivity and specificity were high. However the analysis did not fully take into account the potential endemic nature of multiple flaviviruses in a given region. Consequently the specificity of the responses to the distinct rNS1 proteins would benefit from evaluating the potential reactivity of each NS1 proteins with human serum samples from individuals exposed to different flaviviruses (TBE, WNE, Zika, Dengue, in addition to JE or YF). 

Despite these limitations, the study builds on earlier reports implicating the potential utility of recombinant NS1 proteins to detect exposure to distinct flaviviruses.

Comments

• The authors did not rNS1 from YF or JE in this evaluation. 

• P.5/Line 10-12: Unclear what is meant by “Limited to TBEV, for which a vaccine is available for humans, rNS1 serology has the ability to detect vaccine breakthrough cases of infection”-this methodology is limited to use for TBEV?.

• Figure 1: Panels C,D, and E would benefit from inclusion of molecular weight markers

• Panel C middle panels- indicates “non-denat/reducing” but text p.6/line 19 seems to indicate non denat/non reducing. Please clarify.

• The Last panel- indicates native, would this be equivalent to “non-denat/non-reduced”? Inclusion of molecular markers would be helpful to evaluate “hex” in last panel.

• Figure 2 Panel A- please label to indicate virus per lane and indicate conditions used (eg denat/reduced?)

• Figure 2 Panels C-H: Please clarify which bleed (#day post immunization) was used for the analysis. Please also confirm whether mouse serum was evaluated against rNS1 proteins that were HIS or V5 tagged. 

• P.7/Line 2- Despite characterization of His tagged proteins, immunological / immunogenicity evaluations were performed with V5-tagged. Was there a rationale for this change?

• P.7/Line 5. Immogenicity was evaluated in BALB/c mice following intradermal immunization with plasmid DNA constructs. Was there a reason this approach was utilized rather than performing the evaluation with the purified HIS or V5 proteins? 

• P7/Lines 6-27: The authors are comparing the IgM and IgG response to TBE, WNE, Zika and Den 1-4 NS1 proteins. Most proteins were evaluated following regimen of 1,15,22 and 29. With bleeds on 0,19, 26 and 46. However regimen was changed for Dengue NS1 and mice were immunized on 14 days intervals 1,15,29?(Not 22?) and bleed on days 0,6,22, 34, 57). The difference in the regimens makes it difficult to compare the responses, especially IgM since the early timepoint was only evaluate for the Den NS1 proteins. 

• P7/Line 28: In the specificity testing, it is not clear which bleeds (day) are being evaluated. Please clarify. It would be expected that specificity would increase over time with some reduced specificity temporarily post boost. 

• Page 8/line 22- TBEV diagnosed cases- can you please clarify how these individuals were diagnosed or confirmed (e.g. PCR?).

• General notes: Throughout the text- In many sections of the manuscript the text is not clear when referring to the rNS1-ELISA being evaluated at the time. (example P.9/Line 29-30, “Whereas, 10/16 IgM and 12/16 IgG positives were obtained by rNS1-ELISA”. The text would be easier to follow if the authors could specify the antigen source (eg WNE- rNS1-ELISA)

Reviewer #3: Mora-Cardenas et al describe the expression and perform limited characterisation of recombinant sNS1 from transfected HEK293 cells. Authors then go on to demonstrate the use of NS1 as a target antigen for IgM/IgG serological analysis of vaccinated mice and human samples. The main focus of the article is serological analysis of TBE vaccinated and infected patients with a small number of Zika, dengue and West Nile patients. In attempting to demonstrate the broad utility of NS1 serological analysis the limited number of patients sera tested for dengue, WNV and Zika is a weakness of the manuscript. Furthermore, with the potential importance to disease outcome associated with dengue IgM/IgG serology in terms of primary vs secondary no comment was made regarding the potential value of this assay at identifying primary or secondary dengue. If the authors have access to a larger bank of human samples the manuscript would benefit from an increased sample size.

Provide the calcium phosphate transfection method briefly.

What was the rational for choosing DNA vaccination of mice, given the authors are efficiently expressing protein why not vaccinate with the NS1 antigen itself in ID infection?

PLOS authors have the option to publish the peer review history of their article (what does this mean?). If published, this will include your full peer review and any attached files.

Reviewer #1: No

Reviewer #2: No

Reviewer #3: No

---

## [Decision Letter · Decision Letter 1]

17 Dec 2019

Dear Dr. Marcello:

Thank you very much for submitting your manuscript "Comparative specificity and sensitivity of NS1-based serological assays for the detection of Flavivirus immune response" (PNTD-D-19-01247R1) for review by PLOS Neglected Tropical Diseases. Your manuscript was fully evaluated at the editorial level and by independent peer reviewers. The reviewers appreciated the attention to an important topic but identified some aspects of the manuscript that should be improved.

We therefore ask you to modify the manuscript according to the review recommendations before we can consider your manuscript for acceptance. Your revisions should address the specific points made by each reviewer.

(1) A letter containing a detailed list of your responses to the review comments and a description of the changes you have made in the manuscript.

(2) Two versions of the manuscript: one with either highlights or tracked changes denoting where the text has been changed (uploaded as a "Revised Article with Changes Highlighted" file ); the other a clean version (uploaded as the article file).

(3) If available, a striking still image (a new image if one is available or an existing one from within your manuscript). If your manuscript is accepted for publication, this image may be featured on our website. Images should ideally be high resolution, eye-catching, single panel images; where one is available, please use 'add file' at the time of resubmission and select 'striking image' as the file type. 

Please provide a short caption, including credits, uploaded as a separate "Other" file. If your image is from someone other than yourself, please ensure that the artist has read and agreed to the terms and conditions of the Creative Commons Attribution License at http://journals.plos.org/plosntds/s/content-license (NOTE: we cannot publish copyrighted images). 

(4) Appropriate Figure Files 

Please remove all name and figure # text from your figure files upon submitting your revision. Please also take this time to check that your figures are of high resolution, which will improve both the editorial review process and help expedite your manuscript's publication should it be accepted. Please note that figures must have been originally created at 300dpi or higher. Do not manually increase the resolution of your files. For instructions on how to properly obtain high quality images, please review our Figure Guidelines, with examples at: http://journals.plos.org/plosntds/s/figures

While revising your submission, please upload your figure files to the Preflight Analysis and Conversion Engine (PACE) digital diagnostic tool, https://pacev2.apexcovantage.com/ PACE helps ensure that figures meet PLOS requirements. To use PACE, you must first register as a user. Then, login and navigate to the UPLOAD tab, where you will find detailed instructions on how to use the tool. If you encounter any issues or have any questions when using PACE, please email us at figures@plos.org.

We hope to receive your revised manuscript by Feb 15 2020 11:59PM. If you anticipate any delay in its return, we ask that you let us know the expected resubmission date by replying to this email.

To submit your revised files, please log in to https://www.editorialmanager.com/pntd/

Sincerely,

Sassan Asgari

Guest Editor

A. Desiree LaBeaud

Deputy Editor

Reviewer's Responses to Questions

**Key Review Criteria Required for Acceptance?**

**Methods**

-Are the objectives of the study clearly articulated with a clear testable hypothesis stated?

-Is the study design appropriate to address the stated objectives?

-Is the population clearly described and appropriate for the hypothesis being tested?

-Is the sample size sufficient to ensure adequate power to address the hypothesis being tested?

-Were correct statistical analysis used to support conclusions?

-Are there concerns about ethical or regulatory requirements being met?

Reviewer #2: Revision to the text and supplemental data has improved the manuscript

Reviewer #3: (No Response)

**Results**

-Does the analysis presented match the analysis plan?

-Are the results clearly and completely presented?

-Are the figures (Tables, Images) of sufficient quality for clarity?

Reviewer #2: The last figure in the proofing package (following Supplemental Table 4) includes one labeled :”Striking figure”. It is not clear what is being described in comparison to heat maps in Fig 2 (G,H), and no figure legend is provided.

Reviewer #3: (No Response)

**Conclusions**

-Are the conclusions supported by the data presented?

-Are the limitations of analysis clearly described?

-Do the authors discuss how these data can be helpful to advance our understanding of the topic under study?

-Is public health relevance addressed?

Reviewer #2: The authors do not overstate conclusions, but discuss in terms of potential use.

Reviewer #3: (No Response)

**Editorial and Data Presentation Modifications?**

Reviewer #2: See General Comments

Reviewer #3: (No Response)

**Summary and General Comments**

Reviewer #2: Some remaining questions that could be addressed could include additional information around the specificity of the results when evaluating human samples. Although data is provided regarding cross-reactivity of mouse serum following plasmid immunization, the specificity with human samples is not well characterized. This is of particular importance with regards to samples from individuals post-acute/early convalescent infection, that are likely to have higher levels of broadly reactive antibodies. The limited number of samples is also recognized a potential weakness, but as long as the authors do not overstate conclusions, but discuss in terms of potential there are no concerns.

Reviewer #3: The authors have adequately addressed the suggested changes. I recommend publishing the article.

PLOS authors have the option to publish the peer review history of their article (what does this mean?). If published, this will include your full peer review and any attached files.

Reviewer #2: No

Reviewer #3: No

---

## [Editor Report · Decision Letter 2]

8 Jan 2020

Dear Dr. Marcello,

We are pleased to inform you that your manuscript, "Comparative specificity and sensitivity of NS1-based serological assays for the detection of Flavivirus immune response", has been editorially accepted for publication at PLOS Neglected Tropical Diseases.

Before your manuscript can be formally accepted and sent to production you will need to complete our formatting changes, which you will receive in a follow up email. Please note: your manuscript will not be scheduled for publication until you have made the required changes.

IMPORTANT NOTES

* Copyediting and Author Proofs: To ensure prompt publication, your manuscript will NOT be subject to detailed copyediting and you will NOT receive a typeset proof for review. The corresponding author will have one final opportunity to correct any errors when sent the requests mentioned above. Please review this version of your manuscript for any errors.

* If you or your institution will be preparing press materials for this manuscript, please inform our press team in advance at plosntds@plos.org. If you need to know your paper's publication date for media purposes, you must coordinate with our press team, and your manuscript will remain under a strict press embargo until the publication date and time. PLOS NTDs may choose to issue a press release for your article. If there is anything that the journal should know, please get in touch.

*Now that your manuscript has been provisionally accepted, please log into EM and update your profile. Go to http://www.editorialmanager.com/pntd, log in, and click on the "Update My Information" link at the top of the page. Please update your user information to ensure an efficient production and billing process.

*Note to LaTeX users only - Our staff will ask you to upload a TEX file in addition to the PDF before the paper can be sent to typesetting, so please carefully review our Latex Guidelines [http://www.plosntds.org/static/latexGuidelines.action] in the meantime.

Best regards,

Sassan Asgari

Guest Editor

A. Desiree LaBeaud

Deputy Editor

---

## [Editor Report · Acceptance letter]

21 Jan 2020

Dear Dr. Marcello,

We are delighted to inform you that your manuscript, "Comparative specificity and sensitivity of NS1-based serological assays for the detection of Flavivirus immune response," has been formally accepted for publication in PLOS Neglected Tropical Diseases.

Best regards,

Serap Aksoy

Editor-in-Chief

Shaden Kamhawi

Editor-in-Chief
